# MRI differentiation of usual type endocervical adenocarcinoma and cervical squamous cell carcinoma

Ebru Hasbay[1]*, Sukru Sahin[2], Esra Canan Kelten Talu[3], Muzaffer Sanci[4], Ridvan Binici[1], Ozden Kefeli[1]

**1** Izmir City Hospital, Department of Radiology, Izmir, Turkey, **2** Elazig Fethi Sekin City Hospital, Department of Radiology, Elazig, Turkey, **3** Tepecik Training and Research Hospital, Department of Pathology, Izmir, Turkey, **4** Izmir City Hospital, Department of Gynecological Oncology, Izmir, Turkey

* ebruhasbay@gmail.com

## Abstract

### Objectives

To evaluate the magnetic resonance imaging (MRI) features that may help differentiate usual-type endocervical adenocarcinoma (UEA) from cervical squamous cell carcinoma (SCC), with particular emphasis on diffusion-weighted imaging–derived apparent diffusion coefficient (ADC) values and tumor growth patterns.

### Methods

This retrospective study included 26 patients with histopathologically confirmed UEA and 50 patients with SCC who underwent preoperative pelvic MRI. Quantitative MRI parameters—including tumor size, tumor-to-muscle signal intensity ratios (SIRs) on T1-weighted imaging (T1WI), T2-weighted imaging (T2WI), contrast-enhanced T1-weighted imaging (CE-T1WI), and mean ADC values—were analyzed. Qualitative imaging features such as tumor growth pattern, location, uterine corpus invasion, intratumoral cyst formation, hydrometra/hematometra, and lymphadenopathy were also assessed. Receiver operating characteristic (ROC) analysis was performed to evaluate the diagnostic performance of ADC values. Multivariable logistic regression analysis was used to identify independent predictors of UEA. Interobserver agreement was assessed using intraclass correlation coefficients (ICC) and Cohen's kappa statistics.

### Results

Mean ADC values were significantly higher in the UEA group than in the SCC group ($0.901 \times 10^{-3}$ mm²/s vs. $0.825 \times 10^{-3}$ mm²/s, p = 0.008). ROC analysis showed a statistically significant but moderate discriminatory performance of mean ADC for differentiating UEA from SCC (AUC = 0.659; 95% CI: 0.527–0.790; p = 0.024). Using

**Data availability statement:** All relevant data are within the paper and its Supporting information file.

**Funding:** The author(s) received no specific funding for this work.

**Competing interests:** The authors have declared that no competing interests exist.

**Abbreviations:** MRI, Magnetic Resonance Imaging; UEA, Endocervical Adenocarcinoma of the Usual type; ROC, Receiver-operating characteristic; SIR, Signal Intensity Ratio; T1WI, T1-weighted imaging; T2WI, T2-weighted imaging; CE T1 WI, Contrast Enhanced T1-weighted imaging; ADC, Apparent Diffusion Coefficient; CT, Computed Tomography; FIGO, The International Federation of Gynecologists and Obstetricians; LEEP, Loop Electrosurgical Excision Procedure; HASTE/SSFSE Half-Fourier acquisition single-shot turbo spin echo; FOV, Field of view; TR, Repetition Time; TE, Echo time; DWI, Diffusion-weighted imaging; ssEPI, Single-shot echo planar imaging; FS, Fat Saturated.

the Youden index, an optimal ADC cut-off value of $0.837 \times 10^{-3}$ mm²/s yielded a sensitivity of 65.4% and a specificity of 62.0% for identifying UEA. Endophytic growth was more frequently observed in UEA, whereas mixed growth was more common in SCC (p = 0.023). Intratumoral cyst formation was observed exclusively in UEA (p = 0.004). In multivariable analysis, mean ADC value and tumor growth pattern were independent predictors of UEA. Interobserver agreement for quantitative and qualitative MRI assessments was excellent.

## Conclusion

Higher ADC values, an endophytic growth pattern, and the presence of intratumoral cysts are useful MRI features for differentiating usual-type endocervical adenocarcinoma from cervical squamous cell carcinoma. These imaging characteristics may contribute to improved preoperative diagnostic accuracy and treatment planning.

## Introduction

Cervical cancer is the fourth most common cancer in women, with the fourth highest mortality [1]. Adenocarcinoma (AC) currently accounts for 10–25% of all uterine cervical carcinomas in developed countries [1]. Because of better screening programs, its incidence keeps rising in line with a decline in the incidence of squamous cell carcinoma (SCC) [2]. UEA (Endocervical Adenocarcinoma of Usual type) is the most common type of cervical adenocarcinoma and accounts for approximately 90% of all adenocarcinomas [3].

Some studies have reported that, compared with squamous cell carcinoma (SCC) at the same stage, adenocarcinoma (AC) as a group tends to be more aggressive, less responsive to chemotherapy and radiotherapy, and more likely to metastasize, resulting in lower survival rates and poorer prognosis [4–6]. However, some studies have also found that UEA, a specific subtype of AC, has a prognosis similar to that of SCC [7,8], suggesting that the literature on this issue is not entirely consistent. This distinction highlights the importance of focusing on UEA when evaluating MRI features, as identifying subtype-specific characteristics may improve diagnostic accuracy and guide tailored treatment strategies.

Accurate preoperative diagnosis of UEA is crucial for optimizing treatment planning and improving patient outcomes. Compared to computed tomography (CT), MRI has superior soft tissue contrast, and the sensitivity and accuracy of MRI in diagnosing cervical cancer were higher, as was its detection rate in the diagnosis of stage I and II. Thus, MRI of the pelvis is the preferred imaging modality [9,10].

Although MRI features of cervical adenocarcinoma and SCC have been reported, evidence specifically addressing MRI-based differentiation between usual-type endocervical adenocarcinoma (UEA) and SCC remains limited. Therefore, we aimed to compare the clinical and MRI characteristics of UEA and SCC and to identify imaging findings that may improve preoperative discrimination between these entities.

## Methods

### Ethical approval

The study protocol was approved by the Institutional Review Board (approval number: 2023/12–33, Date: 10/01/2024).

### Patients

All consecutive eligible patients diagnosed between September 2016 and February 2024 who met the inclusion criteria and had preoperative pelvic MRI available were included. The data were accessed for research purposes on January 10, 2024. The data were anonymized, and the researchers had no access to identifiable information. The inclusion criteria were as follows: (i) patients with a clinical diagnosis of International Federation of Gynecology and Obstetrics (FIGO) stage IB– IVB disease; (ii) patients with confirmation of UEA and SCC by histopathological examination after cervical biopsy, conization, Loop Electrosurgical Excision Procedure (LEEP), and hysterectomy; and (iii) patients with records of pelvic MRI. Patients were excluded if they had (i) a FIGO IA (lesion cannot be detected with the imaging) or (ii) poor imaging quality.

Results revealed 176 patients with SCC. Patients without a preoperative MRI (n = 49), those with lesions too small overall to be detected on MRI (n = 62), and those with poor imaging quality (n = 15) were excluded from the study. Also, 46 UEA patients were identified. Those without preoperative MRI (n = 11) and those with lesions too small overall to detect on MRI (n = 7) were all excluded. Thus, the final population consisted of 26 patients with UEA and 50 patients with SCC.

### MRI protocol

MRI was performed using 1.5-T MR scanners (Siemens Avanto, Siemens Aera, and GE Optima360, Erlangen, Germany, and Milwaukee, USA). The imaging protocol included axial, coronal, and sagittal planes using half-Fourier acquisition single-shot turbo spin echo (HASTE/SSFSE) sequences with the following parameters: field of view (FOV) of $420 \times 80$ mm, section thickness of 4 mm with a 20% gap, matrix size of $272 \times 320$, repetition time (TR) of 1200 ms, echo time (TE) of 102 ms, and a scan duration of 50 seconds.

T1-weighted imaging (T1WI) with fat suppression (FS) was performed in the late phase after the intravenous injection of 0.1 mmol/kg gadoterate meglumine (Gd-DOTA, Dotarem; Guerbet, France). Axial diffusion-weighted imaging (DWI) was acquired using single-shot echo planar imaging (ssEPI) with FS and diffusion gradient b-values of 50 and 800 s/mm². Apparent diffusion coefficient (ADC) maps were generated when two or more b-values were available. The scanning range extended from the inferior pubic symphysis to the renal hilum.

ADC measurements were successfully obtained for all patients included in the final analysis.

### Image analysis

All MR imaging was reviewed by experienced radiologists with 14 and 17 years of post-training experience in gynecological imaging who were blinded to the histopathological results. Any discrepancies between the radiologists were resolved by consensus.

As a quantitative assessment, reviewers measured the maximum tumor diameter and signal intensity (SI) of the tumor, by placing regions of interest (ROIs) on the axial T1WI, T2WI, and CE-T1WI. Also, the reviewers measured the SI of the gluteus maximus or iliacus muscles at the level of the tumor to calculate the tumor-to-muscle signal intensity ratio (SIR). Apparent diffusion coefficient (ADC) values of the tumors ($\times 10^{-3}$ mm²/s) were measured on ADC maps by placing ROIs. A circular ROI was carefully positioned on the tumor, as widely as possible in the slice showing the maximum diameter avoiding areas such as hemorrhage and necrosis by referring to other sequences, including T2WI and contrast and non-contrast-enhanced T1WI. At least three measurements were obtained and averaged.

Regarding qualitative assessments, the reviewers evaluated the primary tumor's growth pattern, location, and SI. Three categories were identified for the tumor growth pattern: endophytic growth into the cervical stroma, mixed pattern involving both exophytic and endophytic growth, and exophytic growth from the cervical surface [11] (Fig 1). Three categories were also used to categorize the location of the tumor: the upper half of the cervical canal, the lower half, or the entire cervical canal. In addition to this, SI on CE-T1W images were qualitatively classified into low, moderate, and high relative to the outer myometrium. SI was also classified into low, moderate, and high on T1WI, T2WI, and DWI. The two reviewers classified SI on DWI as high (similar to nerve roots), moderate (similar to the small intestine), or low (similar to the background signal).

Furthermore, the presence of uterine corpus invasion and intratumoral cyst formation, hydrometra or hematometra, abnormal ascites, peritoneal dissemination, lymphadenopathy, and the infiltrative growth pattern were evaluated. One definition of the infiltrative growth pattern was an irregular, poorly recognized interface between the surrounding tissue and the tumor. When a lymph node in the pelvis had a short-axis diameter of more than 8 mm and 10 mm in the upper abdomen, it was considered abnormal [12,13]. Abnormal ascites were evaluated as a positive finding, the presence of ascites exceeding the level of the uterine fundus and/or filling the pelvic cavity [14].

## Statistical analysis

Quantitative variables were assessed for normality using the Shapiro–Wilk test. Continuous variables were compared between groups using the Student's t-test or the Mann–Whitney U test, as appropriate. Categorical variables were compared using the χ² test or Fisher's exact test.

The diagnostic performance of diffusion-weighted imaging (DWI)–derived apparent diffusion coefficient (ADC) values in differentiating usual-type endocervical adenocarcinoma (UEA) from squamous cell carcinoma (SCC) was evaluated using receiver operating characteristic (ROC) curve analysis. The area under the ROC curve (AUC) with 95% confidence intervals (CIs) was calculated. The optimal ADC cut-off value was determined using the Youden index ($J = $ sensitivity + specificity − 1), which identifies the threshold maximizing the combined sensitivity and specificity. Differences between AUCs for ADC and DWI were statistically compared using DeLong's test for two correlated ROC curves.

To identify independent predictors of UEA, a multivariable logistic regression analysis was performed using a backward stepwise selection method (based on Likelihood Ratio). Variables that reached a p-value <0.10 in the univariate analysis (tumor size, growth pattern, and mean ADC) were initially included in the model. This method was preferred to account for potential correlations between predictors and to ensure the most parsimonious model. Odds ratios (ORs) with 95% confidence intervals were calculated. Model fit was assessed using the omnibus likelihood ratio test, and calibration was evaluated using the Hosmer–Lemeshow goodness-of-fit test. A two-sided p value <0.05 was considered statistically significant.

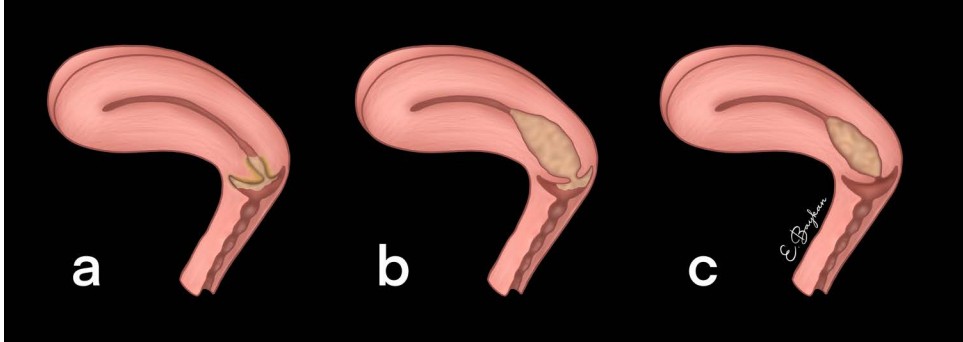

**Fig 1. Illustration of tumor growth pattern.** a. exophytic growth pattern b. mixed growth pattern c. endophytic growth pattern.

Two radiologists independently reviewed all MRI examinations and performed quantitative measurements, including tumor size, ADC values, and tumor-to-muscle signal intensity ratios (SIRs) on T1-weighted imaging (T1WI), T2-weighted imaging (T2WI), and contrast-enhanced T1-weighted imaging (CE-T1WI). Interobserver agreement for quantitative measurements was assessed using the intraclass correlation coefficient (ICC), and the mean values of the two observers were used for subsequent analyses. Agreement between observers for qualitative imaging features was evaluated using Cohen's kappa statistic.

## Results

### Clinical and laboratory characteristics

The clinical and laboratory characteristics of the study population are summarized in Table 1. There were no statistically significant differences between the UEA and SCC groups in terms of age, frequency of elevated tumor markers, menopausal status, or FIGO stage.

The results of the quantitative MRI evaluation are summarized in Table 2. Although the mean tumor size was smaller in the UEA group (41.15 mm) than in the SCC group (50.10 mm), this difference did not reach statistical significance (p = 0.090). Tumor-to-muscle signal intensity ratios (SIRs) on T1-weighted (T1WI) and T2-weighted images (T2WI) were also comparable between groups, with no statistically significant differences.

**Table 1. Clinical Features and Laboratory Findings.**

|  | SCC (n = 50) | UEA (n = 26) | P value |
|---|---|---|---|
| Age (mean±SD) | 55.54 ± 13.51 | 57.81 ± 10.84 | 0.461[a] |
| CA125 (mean±SD) | 83.12 ± 311.26 | 46.36 ± 103.53 | 0.561[b] |
| CEA (mean±SD) | 5.54 ± 8.12 | 6.82 ± 10.61 | 0.562[b] |
| Premenopozal n (%) | 19 (38.0) | 6(23.1) | 0.189[c] |
| Postmenopozal n (%) | 31 (62.0) | 20(76.9) | |
| FIGO (stage 1B,2) | 28(56.0) | 20(76.9) | 0.073[c] |
| FIGO (stage 3,4) | 22(44.0) | 6(23.1) | |

[a]Student's t-test was used.

[b]Mann-Whitney U test was used.

[c]Chi-square test was used.

UEA endocervical adenocarcinoma of the usual type, SCC squamous cell carcinoma.

CA carcinoembryonic antigen, FIGO International Federation of Gynecologists and Obstetricians.

**Table 2. Quantitative Measurements of SCC and UEA.**

|  | SCC (n = 50) | UEA (n = 26) | P value |
|---|---|---|---|
| Tumor size (mm) | 50.10 ± 18.73 | 41.15 ± 26.26 | 0.090 |
| SIR on T1WI | 1.19 ± 0.15 | 1.18 ± 0.21 | 0.841 |
| SIR on T2WI | 3.64 ± 1.07 | 4.03 ± 1.14 | 0.142 |
| SIR on CE-T1WI | 1.72 ± 0.42 | 1.53 ± 0.36 | 0.055 |
| Mean ADC (×10$^{-3}$ mm²/s) | 0.825 ± 0.102 | 0.901 ± 0.140 | 0.008 |

Student's t-test was used.

Values are presented as mean ± standard deviation (SD). Abbreviations: SCC, squamous cell carcinoma; UEA, usual-type endocervical adenocarcinoma; SIR, signal intensity ratio; ADC, apparent diffusion coefficient; T1WI, T1-weighted imaging; T2WI, T2-weighted imaging; CE-T1WI, contrast-enhanced T1-weighted imaging.

The mean contrast-enhanced T1-weighted imaging (CE-T1WI) SIR tended to be lower in UEA than in SCC (1.53 vs 1.72), showing borderline statistical significance (p = 0.055). In contrast, mean apparent diffusion coefficient (ADC) values were significantly higher in UEA compared with SCC (0.901 ± 0.140 vs 0.825 ± 0.102 × 10⁻³ mm²/s), with a mean difference (UEA − SCC) of 0.076 × 10⁻³ mm²/s (95% CI, 0.020–0.132; p = 0.008). The magnitude of this difference corresponded to a moderate effect size (Hedges' g ≈ 0.65).

## ROC analysis

ROC analysis showed moderate discrimination for both parameters in differentiating UEA from SCC. ADC yielded an AUC of 0.659 (95% CI: 0.527–0.790; p = 0.024), as illustrated in Fig 2, with an optimal cut-off of 0.837 × 10⁻³ mm²/s (Youden index), providing 65.4% sensitivity and 62.0% specificity for identifying UEA. DWI demonstrated a slightly higher AUC of 0.714 (95% CI: 0.596–0.832; p = 0.002); the Youden-derived threshold of 2.233 resulted in 80.8% sensitivity and 62.0% specificity. Despite the numerically higher AUC for DWI, DeLong testing indicated no significant difference between the correlated ROC curves (ΔAUC = 0.055; p = 0.592).

## Qualitative MRI evaluation

The qualitative MRI findings are summarized in Table 3. An endophytic growth pattern was more frequently observed in UEA (61.5%) compared with SCC (34%), whereas mixed-type growth patterns were more common in SCC (56%) than in

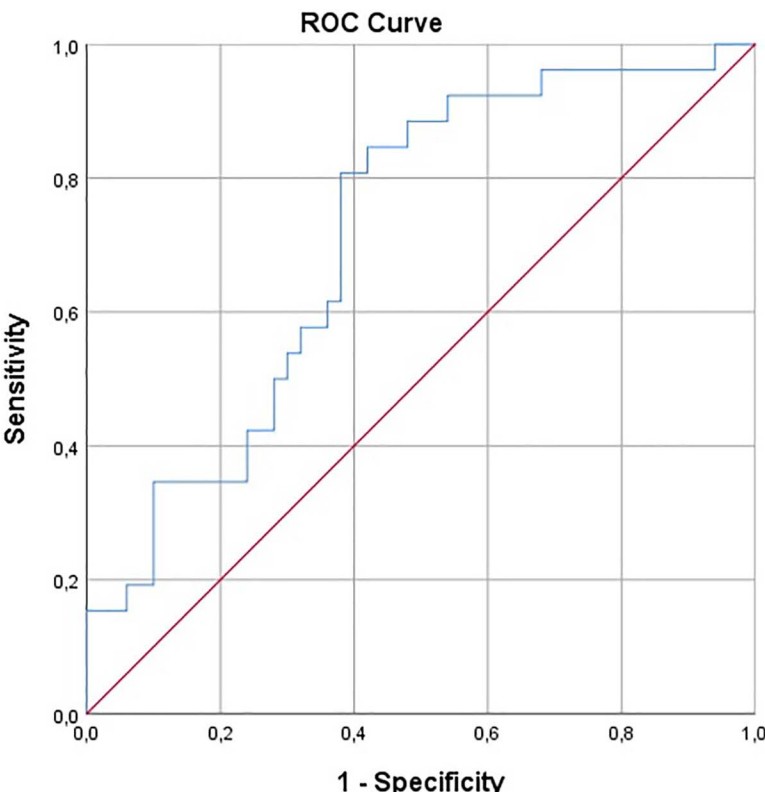

**Fig 2. The receiver operating characteristic (ROC) curve for ADC in UEA compared to SCC.** UEA endocervical adenocarcinoma of the usual type, SCC squamous cell carcinoma.

**Table 3. Qualitative Imaging Findings of SCC and UEA.**

| | SCC (n = 50) | UEA (n = 26) | P value |
|---|---|---|---|
| Infiltrative growth pattern, n (%) | | | 0.516 |
| *present* | 23 (54) | 12 (46.2) | |
| *absent* | 27 (46) | 14 (53.8) | |
| Growth pattern, n (%) | | | 0.023 |
| *endophytic* | 17 (34) | 16 (61.5) | |
| *exophytic* | 5 (10) | 4 (15.5) | |
| *mixed* | 28 (56) | 6 (23.1) | |
| Location, n (%) | | | 0.124 |
| *upper or lower* | 5 (10) | 6 (23.1) | |
| *entire* | 45 (90) | 20 (76.9) | |
| Uterine corpus invasion, n (%) | | | 0.516 |
| *present* | 23 (54) | 14 (53.8) | |
| *absent* | 27 (46) | 12 (46.2) | |
| Intratumoral cyst formation, n (%) | | | 0.004* |
| *present* | 0 (0) | 4 (15.4) | |
| *absent* | 50 (100) | 22 (84.6) | |
| Hydrometra/hematometra, n (%) | | | 0.774* |
| *present* | 9 (18) | 4 (15.4) | |
| *absent* | 41 (82) | 22 (84.6) | |
| Lymphadenopathy, n (%) | | | 0.250 |
| *present* | 18 (36) | 6 (23.1) | |
| *absent* | 32 (64) | 20 (76.9) | |

Chi-square test was used.

UEA endocervical adenocarcinoma of the usual type, SCC squamous cell carcinoma.

*Fisher's exact test was used. Significant differences in values were observed between UEA and SCC (p < 0.05).

UEA (23.1%). Intratumoral cyst formation was not observed in any SCC cases but was detected in 4 patients (17.4%) with UEA. No statistically significant differences were identified for the remaining qualitative MRI features.

The distributions of signal intensity on T1-weighted images, T2-weighted images, diffusion-weighted images, and contrast-enhanced T1-weighted images did not show a significant difference between SCC and UEA (Table 4).

### Interobserver agreement

Interobserver agreement for quantitative measurements was excellent, with intraclass correlation coefficients (ICCs) of 0.984 for tumor size and 0.947 for mean ADC values (Table 5). The ICC for SIR measurements on T1WI was slightly lower but remained acceptable (ICC = 0.803). Agreement between observers for qualitative imaging features—including growth pattern, tumor location, uterine corpus invasion, intratumoral cyst formation, hydrometra/hematometra, and lymphadenopathy—was excellent, with Cohen's κ values ranging from 0.935 to 1.00.

### Multivariable analysis

The multivariable logistic regression model, refined through backward stepwise selection, remained statistically significant (Omnibus $\chi^2$ = 13.35, df = 2, p = 0.001) with adequate calibration (Hosmer–Lemeshow $\chi^2$ = 6.97, df = 8, p = 0.540). The procedure confirmed mean ADC and growth pattern as the independent predictors of UEA. Higher mean ADC values were independently associated with adenocarcinoma (OR = 1.006, 95% CI: 1.002–1.011; p = 0.009). In addition, compared with

**Table 4. Evaluation of MRI Signal Features of SCC and UEA.**

|  | SCC (n = 50) | UEA (n = 26) | P value |
|---|---|---|---|
| SI on T1WI, n (%) |  |  | 0.07* |
| *Low (hypointense)* | 2 (4) | 1 (3.8) |  |
| *Moderate (isointense)* | 48 (96) | 22 (84.6) |  |
| *High (hyperintense)* | 0 (0) | 3 (11.5) |  |
| SI on T2WI, n (%) |  |  | 0.34 |
| *Low (hypointense)* | 0 (0) | 0 (0) |  |
| *Moderate (isointense)* | 0 (0) | 1 (3.8) |  |
| *High (hyperintense)* | 50 (100) | 25 (96.2) |  |
| SI on DWI, n (%) |  |  | 1 |
| *Low (hypointense)* | 0 (0) | 0 (0) |  |
| *Moderate (isointense)* | 2(4) | 1 (3.8) |  |
| *High (hyperintense)* | 48 (96) | 25 (96.2) |  |
| SI on CE-T1WI, n (%) |  |  | 0.11 |
| *Low (hypointense)* | 50 (100) | 24 (92.3) |  |
| *Moderate (isointense)* | 0 (0) | 2 (7.7) |  |
| *High (hyperintense)* | 0 (0) | 0 (0) |  |

UEA endocervical adenocarcinoma of usual type, SCC squamous cell carcinoma.

SI signal intensity, T1WI T1-weighted image, T2WI T2-weighted image, DWI Diffusion weighted image, CE contrast enhanced.

**Table 5. Evaluation of Interobserver Agreement in MRI Measurements with Intraclass Correlation.**

|  | ICC coefficient | 95% CI | P value |
|---|---|---|---|
| Tumor size | 0.984 | 0.975-0.990 | <0.001 |
| ADC value | 0.947 | 0.916-0.966 | <0.001 |
| SIR on T1WI | 0.803 | 0.690-0.875 | <0.001 |
| SIR on T2WI | 0.903 | 0.847-0.939 | <0.001 |
| CE-T1WI | 0.856 | 0.772-0.909 | <0.001 |

ICC intraclass correlation, CI confidence interval.

pure endophytic stromal growth (reference), non-endophytic growth patterns were associated with significantly lower odds of adenocarcinoma (OR = 0.262, 95% CI: 0.090–0.766; p = 0.014).

## Discussion

In our study, mean ADC values were significantly higher in UEA than in SCC ($0.901 \pm 0.140$ vs. $0.825 \pm 0.102 \times 10^{-3}$ mm²/s). ROC analysis demonstrated a statistically significant but moderate discriminatory performance (AUC = 0.659), and the optimal ADC cut-off value was $0.837 \times 10^{-3}$ mm²/s, yielding a sensitivity of 65.4% and a specificity of 62.0% (Fig 3). ADC is considered a quantitative imaging biomarker that reflects tumor microstructure; lower ADC values generally correspond to more restricted diffusion, which is typically related to higher cellularity, reduced extracellular space, and increased proliferative activity [15,16]. Nevertheless, our results indicate that ADC alone provides limited diagnostic accuracy for differentiating UEA from SCC. Although mean ADC was an independent predictor in our multivariable model, its odds ratio (1.006) suggests a small clinical effect size per unit change. Consequently, it should be considered a supportive

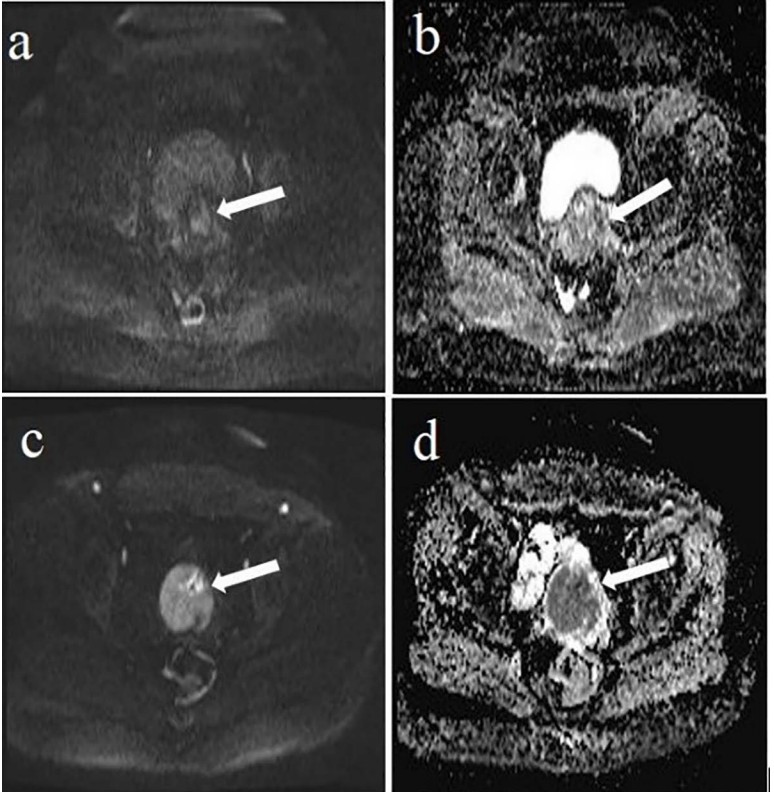

**Fig 3. a. A 42 years old woman with SCC hyperintensity on DWI (arrow). b. Moderate signal intensity on the ADC map, with an ADC value of 1.04×10$^{-3}$ mm²/s (arrow). c. A 48 years old woman with UEA hyperintensity on DWI (arrow). d. Hypointensity on the ADC map, with an ADC value of 0.96x10$^{-3}$ mm²/ s (arrow).**

parameter to be integrated with other MRI features within a multiparametric assessment, rather than a standalone definitive diagnostic tool.

MRI can potentially help differentiate between UEA and SCC based on growth patterns (endophytic, exophytic, and mixed). The squamocolumnar junction is where the majority of cervical squamous cell carcinomas grow. The junction is outside the external uterine os in younger women, and the tumor has an exophytic growth pattern. The junction is typically found in the cervical canal of elderly patients, and the cancer exhibits an endophytic growth pattern, meaning it grows inward along the cervical canal. The endophytic growth pattern is more common in UEA than SCC in our study (Table 3, Figs 4 and 5). However, according to previous studies, approximately 50% of UEA show an invasive growth pattern with an exophytic protrusion, and some exhibit a diffuse infiltrative pattern [11,17,18].

Furthermore, data reported that UEAs were predominantly observed in the exocervix (44.4%) or entire cervix (31.1%) [19]. In our study, UEA has more endophytic growth patterns. The result may be due to UEA arising from glandular cells in the part of the cervix called the endocervical canal. Histopathologically, the diagnosis of invasion by endocervical adenocarcinoma is based on stromal infiltration in the form of marked glandular confluence with cribriform or microacinar architecture. These findings suggest that further studies should be conducted with a larger patient population in this field.

Intratumoral cysts can occasionally be observed in UEA [11]. In our study, intratumoral cysts were more common in UEA, which may be due to their histological features, such as the presence of hemorrhage and necrotic debris within the gland space [17]. The findings mentioned above might indicate that UEAs were more heterogeneous tumors, which is consistent with previous reports [20].

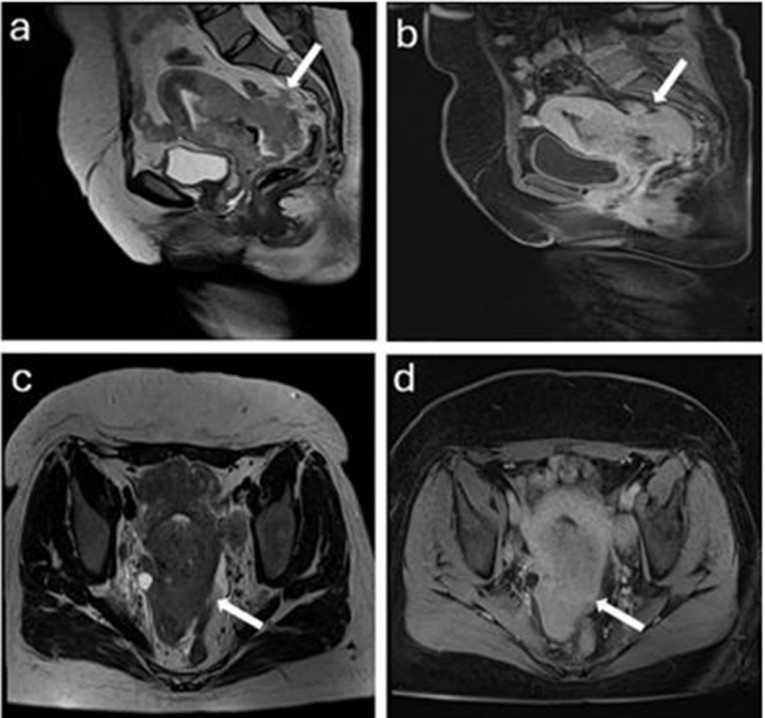

**Fig 4. A 36-year-old woman with SCC.** a. Sagittal T2WI shows a hyperintense lesion relative to the myometrium with invasive growth pattern (arrow) b. Sagittal contrast-enhanced T1WI shows a hypointense lesion relative to the myometrium (arrow). c. Axial T2WI shows a hyperintense lesion relative to the myometrium (arrow). d. Axial contrast-enhanced T1WI shows a hypointense lesion relative to the myometrium (arrow).

In our study, the mean CE ratio in UEA is low compared to SCC, although it was at the border of significance. Histopathologically, UEA consists of neoplastic gland formations with complex architecture and a desmoplastic stromal reaction. These histopathological features, along with the presence of intratumoral cysts, may contribute to this finding. According to the ESUR guidelines, CE-MRI is optional, so it is not used in all centers. Therefore, this finding cannot be used for discrimination alone.

This study has several limitations. First, its retrospective, single-center design may have introduced selection bias. In addition, patients without preoperative MRI or with lesions not detectable on MRI were excluded, which may limit representativeness. Second, because UEA is relatively uncommon, the sample size was modest, potentially reducing statistical power and increasing the risk of overfitting in the multivariable model despite restricting the number of predictors. Third, the findings were not externally validated, and no internal cross-validation was performed, which may limit generalizability. Finally, MRI examinations were acquired over an extended period using different 1.5-T scanners and potentially varying acquisition parameters; this inter-scanner and protocol variability may have affected the comparability of ADC measurements despite excellent interobserver agreement. Therefore, prospective, multicenter studies with standardized imaging protocols and appropriate validation strategies are warranted to confirm and extend our results.

## Conclusion

In conclusion, certain MRI features may help differentiate UEA from SCC. Compared with SCC, UEA demonstrated higher mean ADC values, was more likely to exhibit an endophytic growth pattern, and was more frequently associated with intratumoral cysts. However, given the moderate discriminatory performance of ADC and the retrospective design, these

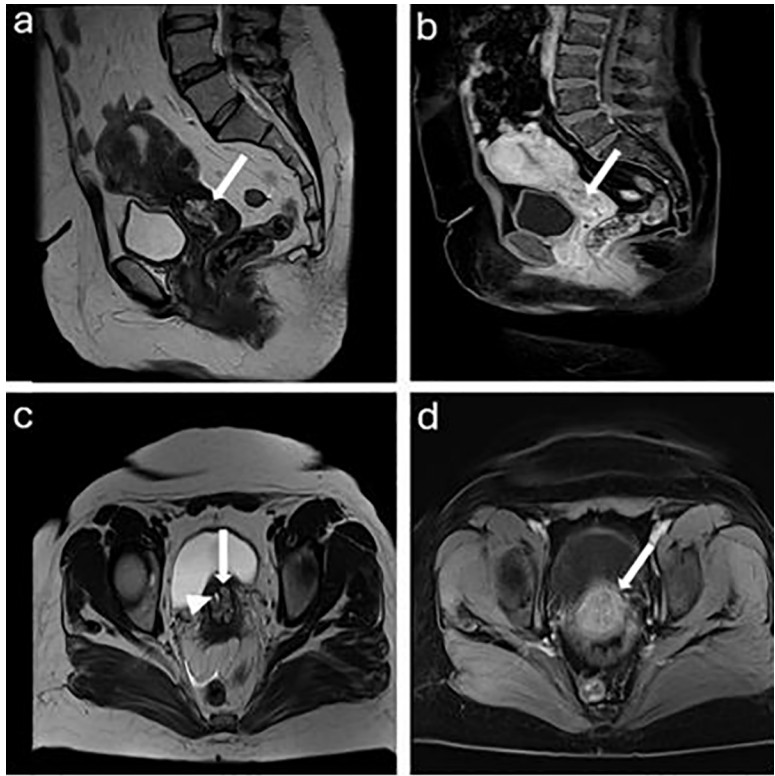

**Fig 5. A 45-years old woman with UEA.** a. Sagittal T2WI shows a relatively hyperintense mass than myometrium with endophytic growth pattern (arrow). b. Sagittal contrast-enhanced T1WI shows a hypointense lesion relative to the myometrium (arrow). c. Axial T2WI shows a UEA (arrow) with intratumoral cyst (arrowhead). d. Axial contrast-enhanced T1WI shows that the lesion appears as slightly hypointense lesion than the myometrium (arrow).

findings should be interpreted as supportive markers within a multiparametric MRI assessment and warrant validation in larger, prospective cohorts.

## Supporting information

**S1 File. Characterization of the SADS-CoV M^pro-Lys35Val mutant.**
(XLSX)

## Author contributions

**Conceptualization:** Ebru Hasbay, Sukru Sahin, Esra Canan Kelten Talu, Muzaffer Sanci.

**Data curation:** Ebru Hasbay.

**Formal analysis:** Sukru Sahin.

**Funding acquisition:** Ebru Hasbay, Sukru Sahin, Esra Canan Kelten Talu, Muzaffer Sanci, Ridvan Binici, Ozden Kefeli.

**Investigation:** Ebru Hasbay.

**Methodology:** Ebru Hasbay, Sukru Sahin.

**Project administration:** Ebru Hasbay.

**Resources:** Ebru Hasbay, Esra Canan Kelten Talu, Muzaffer Sanci.

**Software:** Ebru Hasbay, Ridvan Binici, Ozden Kefeli.

**Supervision:** Ebru Hasbay.

**Validation:** Ebru Hasbay.

**Visualization:** Ebru Hasbay, Sukru Sahin.

**Writing – original draft:** Ebru Hasbay, Sukru Sahin.

**Writing – review & editing:** Ebru Hasbay, Sukru Sahin, Ridvan Binici, Ozden Kefeli.

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
