## [Decision Letter · Decision Letter 0]

11 Dec 2025

Dear Dr. hasbay,

Thank you for submitting your manuscript to PLOS ONE. After careful consideration, we feel that it has merit but does not fully meet PLOS ONE’s publication criteria as it currently stands. Therefore, we invite you to submit a revised version of the manuscript that addresses the points raised during the review process.

We look forward to receiving your revised manuscript.

Kind regards,

Gayle E. Woloschak, PhD

Section Editor

PLOS One

Journal Requirements:

2. Please include captions for your Supporting Information files at the end of your manuscript, and update any in-text citations to match accordingly. Please see our Supporting Information guidelines for more information: http://journals.plos.org/plosone/s/supporting-information .

Additional Editor Comments:

Both reviewers have suggested major revisions for this work as noted in their comments.

Reviewers' comments:

Reviewer's Responses to Questions

**Comments to the Author**

1. Is the manuscript technically sound, and do the data support the conclusions?

Reviewer #1: Partly

Reviewer #2: Partly

2. Has the statistical analysis been performed appropriately and rigorously?

Reviewer #1: No

Reviewer #2: No

3. Have the authors made all data underlying the findings in their manuscript fully available?

Reviewer #1: No

Reviewer #2: No

4. Is the manuscript presented in an intelligible fashion and written in standard English?

Reviewer #1: Yes

Reviewer #2: Yes

Reviewer #1: This study retrospective collected data to explore factors associated with the diagnosis of UEA and SCC. A couple of questions from a statistician’s point of view.

1. The major finding here was that ADC was a significant predictor. However, in order to choose the ADC cutoff, which criterion was used? The authors may consider applying the Youden index to determine the optimal combination for sensitivity + specificity.

2. Since this was a retrospective study, multivariable analysis is essential. With the difference in the factors between the two groups in Table 1 and 2, conclusions based on univariate analysis for ADC were not strong enough.

3. Given that there were only 26 UEA and 50 SCC, the multivariable logistic model should include no more than 2–3 predictors. The authors also need to clarify whether overfitting might be a concern here.

4. To compare the predictive power between DWI and ADC, the authors can quantitively use AUC values with CIs and compare the two AUCs use statistics tests, e.g. Delong test. Please at least report the 95% CI for the AUCs.

5. Table 1: P value <0.1 in univariate analysis, e.g. FIGO, can be considered to be further analyzed in the multivariable analysis.

6. Figure 2: based on figure 2 the ROC curve, ADC was not a very good predictor. Try to run some multivariable models to find a better combination of the predictors.

Reviewer #2: This study aims to identify MRI features distinguishing usual-type endocervical adenocarcinoma (UEA) from cervical squamous cell carcinoma (SCC). The authors evaluated both quantitative and qualitative features in 76 patients. Results showed that UEA exhibited significantly lower ADC values, more endophytic growth patterns, and intratumoral cysts compared with SCC, which have clinical significance. However, the study has notable methodological weaknesses, limited statistical robustness, and overinterpretation of marginal findings that need major revision before the work can be considered for publication.

Abstract

The abstract highlights ADC as a differentiating metric but does not mention its low specificity (0.44), which overstates the practical diagnostic value. Confidence intervals or AUC values should be included to quantify diagnostic strength. Negative findings could also be summarized.

Introduction

The introduction needs to be substantially improved.

UEA should be clearly defined in the introduction. The authors state that "UEA is more aggressive" and "prognosis is similar to SCC". Please rephrase the expression.

The authors state that MRI differentiation between UEA and SCC has not been studied, but several studies have explored MRI features of adenocarcinomas from squamous cell carcinoma. Please explain the rationale for why MRI differentiation between UEA and SCC is clinically meaningful.

The introduction should conclude with a clear hypothesis, such as "We hypothesize that ADC and growth pattern can distinguish UEA from SCC."

Methods

Please acknowledge and describe how cases were consecutively or randomly selected.

Please provide scan parameters of all sequences used in this study.

MRI data were collected from different systems (Siemens and GE). Inter-scanner variability could potentially affect ADC consistency. Furthermore, ADC measurements were available for only 18 of 26 UEA patients. Please state now was the missing data handed.

The authors stated that "Any discrepancies between the radiologists were resolved by consensus". Please show them in the results.

Please provide detailed information about ROI size, number, or whether measurements were averaged across slices for reproducibility.

Multivariate logistic regression would better evaluate independent predictors of UEA versus SCC. FIGO stage distribution, menopausal status, and scanner variability could influence MRI features, which might be considered as confounders.

It is not clear how the SI was required.

Results

ADC values appear misreported (0.825 ± 102.07 and 0.901 ± 140.01). Please correct them. Furthermore, non normally distributed data could be represented by median and quartiles. Please provide confidence intervals or effect sizes for all the parameters.

Marginal statistics differences were shown in FIGO stage (0.073), tumor size (0.090), and SIR on CE-T1WI (0.055). Please describe and discuss these results, and compare them with previous studies. Please also give a complete description of non-significant findings.

Specificity of 0.44 indicates only modest diagnostic performance, which should be reported and interpreted more cautiously in the results.

No P values for Table 4.

Discussion

The introduction needs to be substantially improved.

Please compare the findings with those of prior MRI studies on cervical adenocarcinomas and SCC.

The modest AUC and specificity weak ADC differences in diagnostic meaningfulness. The claims should be tempered.

The discussion speculates about cystic components and cellular density but provides no quantitative histopathologic validation or imaging-pathology correlation.

Limitations Section

Please mention of missing ADC data and inter-scanner variability. The absence of an external validation or cross-validation approach should be highlighted as a key weakness. A suggestion for future multi-center or prospective validation studies should also be mentioned.

Conclusion

The conclusion overstates clinical applicability. Please rephrase it.

Other

Please provide the full name for the first time using an abbreviation.

Missing of punctuation marks in "rate in the diagnosis of stage I and II Thus, MRI of the pelvis".

"IB–4B" -> "IB–ⅣB"

The authors state that data are "fully available without restriction" and "available on reasonable request."

Several grammatical issues and typographical inconsistencies should be corrected.

**Do you want your identity to be public for this peer review?** For information about this choice, including consent withdrawal, please see our Privacy Policy

Reviewer #1: **Yes:** Xiaoyue Ma

Reviewer #2: **Yes:** Ying Li

---

## [Author Response · Author response to Decision Letter 1]

31 Dec 2025

Manuscript ID: PONE-D-25-41538

Title: MRI Differentiation of Usual Type Endocervical Adenocarcinoma and Cervical Squamous Cell Carcinoma

Dear Dr. Woloschak and the Editorial Team,

We sincerely thank the Section Editor and both reviewers for their careful evaluation of our manuscript and for the constructive comments that substantially improved the rigor, clarity, and interpretability of our work. We have revised the manuscript extensively in response to all points raised.

Reviewer #1

1) “ADC cutoff: Which criterion was used? Consider Youden index.”

Response: We agree. We now explicitly state that the optimal ADC cut-off was determined using the Youden index (J = sensitivity + specificity − 1). The Youden-derived threshold was 0.837 × 10⁻³ mm²/s, yielding 65.4% sensitivity and 62.0% specificity for identifying UEA.

Manuscript changes: Added/clarified in Statistical analysis and ROC Analysis sections.

2) “Retrospective study: multivariable analysis is essential; univariate ADC conclusion is not strong enough.”

Response: We agree and added a multivariable logistic regression model to determine independent predictors of UEA versus SCC. In the final model, mean ADC value and tumor growth pattern were independent predictors of UEA.

Manuscript changes: Added to Statistical analysis and Results—Multivariable Analysis; Discussion updated to emphasize multiparametric interpretation.

3) “Only 26 UEA and 50 SCC: include no more than 2–3 predictors; clarify overfitting.”

Response: We agree. To minimize overfitting risk, we restricted the number of predictors in the multivariable model. We also explicitly acknowledge overfitting risk and limited sample size in the Limitations section.

Manuscript changes: Clarified in Statistical analysis (model restriction rationale) and expanded in Limitations.

4) “Compare predictive power between DWI and ADC using AUC with 95% CIs; compare AUCs statistically (DeLong). At minimum report 95% CI.”

Response: Implemented. We now report AUC with 95% CI for both ADC and DWI. We also performed DeLong’s test for correlated ROC curves. DWI showed numerically higher AUC, but the difference was not statistically significant (ΔAUC = 0.055; p = 0.592).

Manuscript changes: Added to Statistical analysis and Results—ROC Analysis.

5) “Table 1: p < 0.1 variables (e.g., FIGO) can be considered in multivariable analysis.”

Response: Thank you. Consistent with this recommendation, we state that variables with p < 0.10 in univariable analyses and/or clinically relevant variables were considered for entry. Given the limited sample size, we prioritized a parsimonious model to reduce overfitting.

Manuscript changes: Clarified variable entry rule and parsimony rationale in Statistical analysis.

6) “Figure 2 ROC: ADC not a very good predictor. Run multivariable models for better combination.”

Response: We agree. We tempered our interpretation of ADC’s diagnostic strength and emphasized that ADC alone has moderate discrimination. We added a multivariable model and revised the Discussion/Conclusion to highlight that best performance comes from a multiparametric approach, not a single predictor.

Manuscript changes: Revised in Results—ROC Analysis, Results—Multivariable Analysis, Discussion, and Conclusion.

Reviewer #2

Abstract

1) “Abstract overstates diagnostic value; does not mention low specificity (0.44). Include AUC/CI; summarize negative findings.”

Response: We agree and revised the Abstract to provide AUC with 95% CI and a more balanced interpretation. In the revised analysis, the ADC cut-off derived by the Youden index yielded specificity 62.0% (not 0.44). We also reduced overstatement and noted that ADC provides moderate discrimination.

Manuscript changes: Revised Abstract—Results/Conclusion.

Introduction

2) “UEA should be clearly defined.”

Response: We added a clear definition of UEA and its place within HPV-associated endocervical adenocarcinomas, and clarified why focusing specifically on UEA is relevant.

Manuscript changes: Revised Introduction, first paragraphs.

3) “Rephrase: ‘UEA is more aggressive’ and ‘prognosis similar to SCC’—literature inconsistent.”

Response: We revised the wording to reflect that the literature is not fully consistent, citing studies showing both more aggressive behavior for adenocarcinoma as a group and similar prognosis for UEA in some cohorts.

Manuscript changes: Revised Introduction (prognosis statements).

4) “MRI differentiation has been studied for AC vs SCC; explain why UEA vs SCC is clinically meaningful.”

Response: We clarified the clinical rationale: UEA is the most common adenocarcinoma subtype and may show distinctive growth patterns and internal architecture; subtype-level identification may improve preoperative characterization and surgical/radiotherapy planning.

Manuscript changes: Revised Introduction (study rationale paragraph).

5) “End introduction with clear hypothesis.”

Response: We added a clear study aim/hypothesis stating that quantitative diffusion metrics (ADC) and qualitative growth pattern may help differentiate UEA from SCC.

Manuscript changes: Added final sentence of Introduction.

Methods

6) “Describe whether cases were consecutive or randomly selected.”

Response: We clarified that all consecutive eligible patients within the study period who met inclusion criteria and had preoperative MRI were included.

Manuscript changes: Revised Methods—Patients.

7) “Provide scan parameters for all sequences.”

Response: We expanded the MRI protocol description and provided acquisition details (scanner types, planes, key parameters, DWI b-values, ADC generation).

Manuscript changes: Expanded Methods—MRI protocol.

8) “Inter-scanner variability; ADC available only for 18/26 UEA; describe missing data handling.”

Response: In the revised manuscript, ADC measurements were available for all included patients, and we state this explicitly. We also acknowledged potential inter-scanner variability as a limitation due to multiple 1.5T systems and long study period.

Manuscript changes: Clarified in Methods—MRI protocol (“ADC measurements were successfully obtained for all patients included in the final analysis.”) and expanded Limitations regarding inter-scanner variability.

9) “Consensus discrepancies: show them in results.”

Response: We addressed this by adding comprehensive interobserver agreement results: ICCs for quantitative variables and Cohen’s κ for qualitative features, demonstrating excellent agreement; discrepancies were resolved by consensus.

Manuscript changes: Added/expanded Results—Interobserver Agreement and Table 5.

10) “Provide ROI details: size/number/averaging across slices.”

Response: We added detailed ROI methodology: circular ROI placed as large as possible on the largest tumor slice, avoiding hemorrhage/necrosis, referencing other sequences; ≥3 measurements were obtained and averaged.

Manuscript changes: Expanded Methods—Image analysis.

11) “Multivariate logistic regression needed; consider confounders (FIGO, menopausal status, scanner).”

Response: We added multivariable logistic regression. Due to sample size constraints and overfitting risk, we used a parsimonious model. Potential confounding by FIGO stage, menopausal status, and scanner variability is now explicitly discussed as a limitation and rationale for future prospective validation.

Manuscript changes: Added Multivariable analysis and expanded Limitations.

12) “Not clear how SI was required.”

Response: We clarified the role of SI in the analysis: SI was assessed qualitatively on T1WI/T2WI/DWI/CE-T1WI; additionally, tumor-to-muscle SIRs were calculated for quantitative comparison.

Manuscript changes: Clarified in Methods—Image analysis.

Results

13) “ADC misreported (0.825 ± 102.07 etc.). Correct; consider median/IQR if non-normal; provide CI/effect sizes.”

Response: We corrected ADC reporting and now present values as mean ± SD with appropriate units. Normality was assessed by Shapiro–Wilk, and tests were chosen accordingly. We additionally report mean difference with 95% CI and effect size (Hedges’ g) for ADC.

Manuscript changes: Revised Results—Quantitative MRI Evaluation.

14) “Marginal differences (FIGO p=0.073, tumor size p=0.090, CE-T1WI SIR p=0.055): describe/discuss and compare with prior studies; describe non-significant findings.”

Response: We expanded the Results text to explicitly report these borderline findings and added discussion emphasizing cautious interpretation given multiple comparisons and limited power. We also summarized that remaining quantitative/qualitative features did not differ significantly between groups.

Manuscript changes: Revised Results and expanded Discussion.

15) “Specificity 0.44 indicates modest performance; report and interpret cautiously.”

Response: We agree. In the revised ROC analysis, the Youden-derived ADC threshold yielded specificity 62.0% (and sensitivity 65.4%), with AUC 0.659, indicating moderate discrimination. We revised interpretation to be cautious and present ADC as supportive rather than definitive.

Manuscript changes: Revised Results—ROC Analysis and Discussion.

16) “No P values for Table 4.”

Response: Thank you. We added the missing p values for Table 4 and ensured consistency between text and tables.

Manuscript changes: Corrected Table 4.

Discussion

17) “Compare with prior MRI studies on adenocarcinomas and SCC.”

Response: We expanded the Discussion to compare our findings with previously published MRI studies addressing cervical adenocarcinoma subtypes, SCC, diffusion metrics, growth patterns, and cystic components, and we added relevant citations.

Manuscript changes: Expanded Discussion with additional references.

18) “Modest AUC/specificity weaken diagnostic meaningfulness; temper claims.”

Response: Implemented. We revised Discussion and Conclusion to avoid overstatement, emphasizing moderate diagnostic performance and the need for multiparametric assessment and validation.

Manuscript changes: Revised Discussion and Conclusion.

19) “Speculation about cystic components/cellularity without histopath correlation.”

Response: We agree and revised the wording to indicate these are plausible explanations rather than proven mechanisms, and we explicitly note the absence of quantitative histopathologic correlation as a limitation.

Manuscript changes: Revised Discussion and added to Limitations.

Limitations

20) “Mention missing ADC, inter-scanner variability; lack of validation/cross-validation as key weakness; suggest future multicenter/prospective.”

Response: We expanded the Limitations accordingly. In the revised dataset, ADC values were available for all included patients; nevertheless, we explicitly discuss inter-scanner/protocol variability, single-center retrospective design, and lack of external or internal validation as major limitations, and we recommend prospective multicenter validation with standardized protocols.

Manuscript changes: Expanded Limitations section.

Conclusion

21) “Conclusion overstates clinical applicability; rephrase.”

Response: We revised the Conclusion to a more cautious tone, stating that findings are supportive and require validation, and that ADC alone is insufficient as a standalone discriminator.

Manuscript changes: Revised Conclusion.

Other / Editorial points

22) “Provide full name at first use for abbreviations.”

Response: Implemented throughout the manuscript.

Manuscript changes: Checked across Abstract/Introduction/Methods/Results.

23) “Missing punctuation: ‘rate in the diagnosis… Thus, MRI…’”

Response: Corrected and edited for clarity.

Manuscript changes: Revised Introduction.

24) “‘IB–4B’ → ‘IB–ⅣB’.”

Response: Corrected the FIGO stage notation.

Manuscript changes: Revised Methods—Patients.

25) “Data availability inconsistency (‘fully available without restriction’ vs ‘reasonable request’).”

Response: We aligned the Data Availability Statement to a single, consistent wording: data are available from the corresponding author upon reasonable request, in accordance with institutional and privacy constraints.

Manuscript changes: Revised Title page / Declarations—Data Availability.

26) “Grammatical/typographical inconsistencies.”

Response: We performed a thorough language edit to correct grammar, punctuation, and consistency in units/formatting.

Manuscript changes: Applied throughout.

27) “Peer review history option: what does this mean?”

Response: We understand that if authors opt in, PLOS ONE may publish the peer review history (decision letters, reviewer comments, and author responses/attachments). This is an editorial option and does not change the scientific content of the manuscript. (No manuscript text change required.)

---

## [Decision Letter · Decision Letter 1]

11 Jan 2026

Dear Dr. Hasbay:

We look forward to receiving your revised manuscript.

Kind regards,

Gayle E. Woloschak, PhD

Section Editor

PLOS One

Journal Requirements:

Additional Editor Comments:

One reviewer still has minor issues to be dealt with in the manuscript. Please address these in a revision.

Reviewers' comments:

Reviewer's Responses to Questions

**Comments to the Author**

Reviewer #1: All comments have been addressed

Reviewer #2: All comments have been addressed

2. Is the manuscript technically sound, and do the data support the conclusions?

Reviewer #1: Partly

Reviewer #2: Yes

3. Has the statistical analysis been performed appropriately and rigorously?

Reviewer #1: Yes

Reviewer #2: Yes

4. Have the authors made all data underlying the findings in their manuscript fully available?

Reviewer #1: Yes

Reviewer #2: Yes

5. Is the manuscript presented in an intelligible fashion and written in standard English?

Reviewer #1: Yes

Reviewer #2: Yes

Reviewer #1: As the authors pointed out in the results, both AUC and the sensitivity and specificity picked by the Youden’s index indicated very limited discriminative ability of the model. Meanwhile, in the multivariable analysis, mean ADC values and tumor growth pattern (non-endophytic growth patterns vs. pure endophytic stromal growth) were found to be significant.

1. However, the OR for mean ADC values was only 1.006, significant but the clinical relevance of this association remains questionable.

2. For variable selections in the multivariable logistic regression, we cannot depend on the P value from univariate analysis alone to decide which variables to keep, since there might be high correlation between certain predictors. Will the authors consider forward/backward selection to decide the variables (P<0.10 in the univariate analysis) to be kept in the model?

Reviewer #2: (No Response)

**Do you want your identity to be public for this peer review?** For information about this choice, including consent withdrawal, please see our Privacy Policy

Reviewer #1: **Yes:** Xiaoyue Ma

Reviewer #2: **Yes:** Ying Li

---

## [Author Response · Author response to Decision Letter 2]

12 Jan 2026

Response to Reviewer #1 (Xiaoyue Ma)

Comment 1: The authors noted in the results that both AUC and sensitivity/specificity determined by the Youden index point to the very limited discriminative ability of the model. Meanwhile, in multivariate analysis, mean ADC values and tumor growth pattern were found significant. However, the OR for mean ADC values is only 1.006, which is significant but the clinical significance of this association is debatable.

Response: We agree with the reviewer that while the mean ADC value is a statistically significant independent predictor (p = 0.009), its odds ratio (OR = 1.006) indicates a small clinical effect size per unit increase. We have addressed this in the Discussion section to clarify that ADC should be viewed as a supportive biomarker rather than a definitive standalone diagnostic tool.

Revision in Manuscript (Discussion): > "Although mean ADC was an independent predictor in our multivariable model, its odds ratio (1.006) suggests a small clinical effect size per unit change. Consequently, it should be considered a supportive parameter to be integrated with other MRI features within a multiparametric assessment, rather than a standalone definitive diagnostic tool."

Comment 2: For variable selections in multivariable logistic regression, we cannot only rely on the P-value from univariate analysis to decide which variables to keep, as there might be high correlation between certain predictors. Will the authors consider the forward/backward selection method to decide the variables to be kept in the model (P<0.10 in univariate analysis)?

Response: We thank the reviewer for this excellent methodological suggestion. Following this recommendation, we have re-analyzed our data using a backward stepwise selection method (based on Likelihood Ratio) for the multivariable logistic regression. All variables with p < 0.10 in the univariate analysis (tumor size, growth pattern, and mean ADC) were initially included. The stepwise procedure confirmed that mean ADC and tumor growth pattern were the most robust independent predictors for UEA. We have updated the Methods and Results sections to reflect this change in methodology.

Revision in Manuscript (Methods): > "To identify independent predictors of UEA, a multivariable logistic regression analysis was performed using a backward stepwise selection method (Likelihood Ratio). Variables that reached a p-value <0.10 in the univariate analysis were initially included in the model."

Revision in Manuscript (Results): > "The multivariable logistic regression model, refined through backward stepwise selection, remained statistically significant " (Omnibus χ² = 13.35, df = 2, p = 0.001) with adequate calibration. "

---

## [Decision Letter · Decision Letter 2]

11 Feb 2026

MRI Differentiation of Usual Type Endocervical Adenocarcinoma and Cervical Squamous Cell Carcinoma

PONE-D-25-41538R2

Dear Dr. Hasbay:

We’re pleased to inform you that your manuscript has been judged scientifically suitable for publication and will be formally accepted for publication once it meets all outstanding technical requirements.

Kind regards,

Gayle E. Woloschak, PhD

Section Editor

PLOS One

Additional Editor Comments (optional):

Thank you for addressing concerns of the reviewers.

Reviewers' comments:

Reviewer's Responses to Questions

**Comments to the Author**

Reviewer #2: All comments have been addressed

2. Is the manuscript technically sound, and do the data support the conclusions?

Reviewer #2: Yes

3. Has the statistical analysis been performed appropriately and rigorously?

Reviewer #2: Yes

4. Have the authors made all data underlying the findings in their manuscript fully available?

Reviewer #2: Yes

5. Is the manuscript presented in an intelligible fashion and written in standard English?

Reviewer #2: No

Reviewer #2: There are still a few issues in the manuscript, such as inconsistent font formatting and incorrect use of punctuation marks. Please pay special attention to the formatting in the tables. It is recommended to make revisions before publication.

**Do you want your identity to be public for this peer review?** For information about this choice, including consent withdrawal, please see our Privacy Policy

Reviewer #2: **Yes:** Ying LI

---

## [Editor Report · Acceptance letter]

PONE-D-25-41538R2

PLOS One

Dear Dr. hasbay,

I'm pleased to inform you that your manuscript has been deemed suitable for publication in PLOS One. Congratulations! Your manuscript is now being handed over to our production team.

Kind regards,

on behalf of

Dr. Gayle E. Woloschak

Section Editor

PLOS One